# Efficacy and Safety of Tacrolimus as Treatment for Bleeding Caused by Hereditary Hemorrhagic Telangiectasia: An Open-Label, Pilot Study

**DOI:** 10.3390/jcm11185280

**Published:** 2022-09-07

**Authors:** Josefien Hessels, Steven Kroon, Sanne Boerman, Rik C. Nelissen, Jan C. Grutters, Repke J. Snijder, Franck Lebrin, Marco C. Post, Christine L. Mummery, Johannes-Jurgen Mager

**Affiliations:** 1Department of Pulmonology, St. Antonius Hospital, Koekoekslaan 1, 3435 CM Nieuwegein, The Netherlands; 2Department of Otorhinolaryngology, St. Antonius Hospital, Koekoekslaan 1, 3435 CM Nieuwegein, The Netherlands; 3Department of Pulmonology, University Medical Center Utrecht, Heidelberglaan 100, 3484 CX Utrecht, The Netherlands; 4Department of Internal Medicine, Leiden University Medical Center, Albinusdreef 2, 2333 ZA Leiden, The Netherlands; 5Department of Cardiology, University Medical Center Utrecht, Heidelberglaan 100, 3484 CX Utrecht, The Netherlands or; 6Department of Cardiology, St. Antonius Hospital, Koekoekslaan 1, 3435 CM Nieuwegein, The Netherlands; 7Department of Anatomy and Embryology, Leiden University Medical Center, Albinusdreef 2, 2333 ZA Leiden, The Netherlands

**Keywords:** anemia, epistaxis, gastrointestinal hemorrhage, telangiectasia, hereditary hemorrhagic, tacrolimus

## Abstract

Haploinsufficiency for Endoglin (ENG) and activin A receptor type II-like I (*ACVRL1*/ALK1) lead to the formation of weak and abnormal vessels in hereditary hemorrhagic telangiectasia (HHT). These cause epistaxis (nosebleeds) and/or gastrointestinal blood loss. In vitro in cultured endothelial cells, tacrolimus has been shown to increase ENG and ALK1 expression. It is, therefore, a potential treatment option. We report here a proof-of-concept study in patients with HHT and severe epistaxis and/or gastrointestinal bleeding who were treated daily with orally-administered tacrolimus for twenty weeks. Twenty-five patients with HHT (11 females (44%)) and median age of 59 years were enrolled. Five patients (20%) stopped the trial prematurely-four due to (serious) adverse events ((S)AE). Twenty patients were included in further analyses. Hemoglobin levels increased during tacrolimus treatment from 6.1 (IQR 5.2–6.9) mmol/L at baseline (9.8 g/dL) to 6.7 (6.5–7.1) mmol/L (10.8 g/dL), *p* = 0.003. The number of blood transfusions over the twenty weeks decreased from a mean of 5.0 (±9.2) to 1.9 (±3.5), *p* = 0.04. In 64% of the patients, at least one AE occurred. Oral tacrolimus, thus, significantly increased hemoglobin levels and decreased blood transfusion needs, epistaxis and/or gastrointestinal bleeding in patients with HHT. However, side-effects were common. Further investigation of the potential therapeutic benefit is justified by the outcome of the study.

## 1. Introduction

Hereditary Hemorrhagic Telangiectasia (HHT) is an autosomal dominant, inherited disease, most commonly caused by mutations in the genes encoding endoglin (ENG) or activin A receptor type II-like 1 (*ACVRL1*, or ALK1). These are termed HHT type 1 and HHT type 2, respectively [1,2]. The genes encode proteins in the transforming growth factor β signaling pathway. It is believed that haploinsufficiency is an important cause of the pathogenicity in HHT. An imbalance in activating and inhibiting angiogenic stimuli results in the development of abnormal vasculature that can easily rupture [3]. Most patients with HHT suffer from spontaneous and recurrent epistaxis and/or gastrointestinal bleeding. Despite several treatment modalities that include systemic medical options and invasive surgery, bleeding can still be difficult to control. This leads to iron-deficiency anemia that may require intravenous iron treatment or blood transfusions [4]. Moreover, the quality of life (QoL) is often reduced and fatigue increased by the severity of the bleeding in patients with HHT [5].

Tacrolimus is a widely used immunosuppressive drug for preventing organ rejection in transplant recipients. The first reported evidence of tacrolimus as a potential treatment for HHT-related bleeding was published in 2006 [6]: after a liver transplant in an HHT patient, several drugs, including tacrolimus and sirolimus, surprisingly led to decreased bleeding severity. In two in vitro studies with human endothelial cells and an HHT mouse model, the therapeutic potential of tacrolimus was further studied. Tacrolimus increased ENG and ALK1 expression and decreased the formation of abnormal vasculature [7,8]. Clinical data on the effects of tacrolimus in HHT are limited: currently, only three case reports have been published [6,9,10]. These noted a substantial decrease in bleeding after initiation of tacrolimus treatment. In another clinical trial in which tacrolimus was delivered as a nasal ointment, epistaxis severity was significantly decreased in both the tacrolimus treated- and placebo groups compared to baseline but there was a significant difference between the groups [11]. However, there was no significant difference between the groups when comparing the epistaxis severity before and six weeks after treatment. To date, no clinical trials investigating the potential of oral tacrolimus to treat HHT-related bleeding have been reported. Based on the evidence suggesting oral tacrolimus might reduce epistaxis and gastrointestinal bleeding in patients with HHT, its low cost and its reasonably good tolerability, we designed a non-randomized, open-label, pilot study to investigate its therapeutic potential in HHT-related epistaxis and gastrointestinal bleeding.

## 2. Materials and Methods

### 2.1. Study Design

The study was designed as an open-label, prospective, non-randomized, phase 2, single-center study (European Union Drug Regulating Authorities Clinical Trials (EudraCT) number 2019-003585-40). It was performed in the St. Antonius Hospital, the Dutch HHT expertise center. The study was approved by the ethics committee of the St. Antonius Hospital (MEC-U) (NL71405.100.19/ R19.071) on 17 December 2019. All patients signed informed consent for inclusion before participation in the study. This study is in accordance with the Declaration of Helsinki. The study was started in March 2020 and finished in December 2021.

### 2.2. Patient Selection

Patients were accepted in the trial if they were 18 years or older at the time of inclusion and had been diagnosed with HHT (genetically confirmed diagnosis or ≥3 positive Curacao criteria) [12]. The presence of anemia (sex-dependent; hemoglobin level of <7.3 mmol/L (11.8 g/dL) for females and <7.8 mmol/L (12.6 g/dL) for males), iron deficiency (ferritin level < 10 µg/L), iron treatment or blood transfusions in the last six months were criteria for inclusion. Patients also needed to report 4 or more episodes of epistaxis per week and/or have documented gastrointestinal telangiectases by endoscopy with suspicion of bleeding (e.g., melena, anemia disproportional to epistaxis). In addition, evidence of failure or partial failure of local treatment was required. The reason for inclusion in the tacrolimus trial was determined before enrollment: (1) mainly for epistaxis (no history or clinical evidence of gastrointestinal bleeding and anemia, if present, proportional to epistaxis severity); (2) mainly for gastrointestinal bleeding (no, or only very mild, epistaxis) or (3) a combination of both epistaxis and gastrointestinal bleeding. Patients were excluded if they were suffering from a severe disease with a life expectancy of <1 year, had been diagnosed with severe kidney disease, had a history of severe ventricular cardiac dysfunction, were taking drugs that are contraindicated with tacrolimus, were pregnant, nursing, had a pregnancy wish in the study period or used anticonception inadequately, were currently receiving chemotherapy, or had known hypersensitivity or were allergic to tacrolimus, could not give adequate informed consent or were without sufficient understanding of the Dutch or English language.

### 2.3. Treatment Plan

Patients visited the St. Antonius hospital at trial inclusion (screening visit), after four weeks (baseline visit), eight weeks and at the end of the trial (24 weeks) (see Figure 1). At weeks 12, 16 and 20, follow-up was done by telephone. In case of unexpected events, additional visits were planned as needed. Patients were instructed to record their nosebleeds in an epistaxis diary during the four weeks from the screening visit to the baseline visit. After the baseline visit, tacrolimus (Advagraf^®^ MGA, Astellas Pharma Europe BV, Leiden, The Netherlands) was started once daily for 20 weeks. The starting dose of tacrolimus was 1 mg, and, after approximately one week of treatment, the trough level of tacrolimus was determined. Subsequently, the tacrolimus dose was adjusted if needed: tacrolimus was increased or decreased in steps of 0.5 or 1 mg or continued at the same dose. The target value (trough level) chosen was between 2 and 3 μg/L of tacrolimus. This was based on two case reports with tacrolimus in patients with HHT (Sommer et al., 2019: 1.5–2.5 μg/L [9] and Pruijsen et al., 2021: 2–4 μg/L [10]) and our own clinical experience with 2 patients who had been treated with tacrolimus for HHT-related symptoms prior to this trial. Of note, these trough levels are substantially lower than the target levels in solid organ transplant recipients.

Trough levels were usually determined approximately one week after adjusting the tacrolimus dose. In the case of a stable level within the target range, the period between two trough level measurements was expanded gradually, eventually to at least every 4 weeks. Patients had blood taken at their local hospital and subsequently the blood was sent by special post to our hospital for tacrolimus level measurements. The lowest measurable level of tacrolimus was 1 μg/L. Before the start of treatment, and every four weeks thereafter, lab testing was performed which included measurement of renal function, electrolytes, glucose and complete blood count. Before the start of treatment, liver function and albumin were measured as well.

All patients continued to receive standard of care during the trial for epistaxis and gastrointestinal blood loss i.e., similar to that prior to enrollment and including red blood cell (RBC) units, intravenous iron infusions and invasive surgical interventions. The patients received RBC units and intravenous iron infusions in their own hospital and the need was determined by the patient’s independent treating physician, based on symptoms and hemoglobin levels below approximately 4–5 mmol/L (6.4–8.1 g/dL). The independent treating physician was informed about participation in this trial but was not required to follow a standardized management plan.

### 2.4. Outcomes

The primary outcome was the change in hemoglobin levels measured at the end of the trial (week 24) compared to the baseline visit (week 4). Secondary outcomes included safety and side effects, the change in ferritin levels and the Epistaxis Severity Score (ESS) [13] at the end of the trial compared to baseline. Epistaxis severity was further assessed with an epistaxis diary in which patients recorded their epistaxis events in two periods: before treatment (the first four weeks of the trial) and during treatment (the last four weeks of the trial). The number and duration of the epistaxis episodes as recorded in the diary during and before treatment were compared. Furthermore, the number of RBC transfusions and iron infusions during the study period (20 weeks) were compared with the number of transfusions and infusions required to treat the anemia in the 20 weeks before treatment. Moreover, the quality of life (QoL) measured with the Short Form Health Survey 36 (SF-36) and the level of fatigue measured with Multidimensional Fatigue Inventory 20 (MFI-20) were assessed [14,15]. Lastly, patients were questioned on their satisfaction with the treatment and their willingness to repeat it, both on a five-point scale.

### 2.5. Statistical Analysis

Quantitative data were presented as absolute frequencies or percentages. The median and interquartile range (IQR) were used and subsequently, non-parametric testing was performed due to the small sample size. We used the Wilcoxon signed-rank test for continuous and ordinal parameters to assess the differences at baseline compared to the end of treatment. We chose to report the number of iron infusions and blood transfusions with both the median and interquartile range and the mean and standard deviation (SD); both non-parametric (Wilcoxon signed-rank test) and parametric tests (paired sample T-test) were used to assess differences before enrollment and during the trial. This was because of experiences in a previous trial where not all patients would require intravenous iron treatment or RBC transfusions, making median and interquartile range (IQR) values more difficult to interpret. A *p*-value < 0.05 was regarded as statistically significant. Statistical analysis was performed with SPSS version 24.0 for Windows (IBM, Armonk, NY, USA) and GraphPad 5 (GraphPad Software, Inc., La Jolla, CA, USA).

## 3. Results

### 3.1. Baseline Characteristics

During this trial, 25 patients were included of whom 11 (44%) were female. The median age was 59 (IQR 52–66). The indication for enrollment was divided into three categories: patients suffering from mainly epistaxis (*n* = 16 (64%)), patients suffering mainly gastrointestinal bleeding without clinically relevant epistaxis (*n* = 2 (8%)) and patients suffering both epistaxis and gastrointestinal bleeding (*n* = 7 (28%)). All patients were clinically diagnosed with HHT according to the Curacao criteria. A total of 13 patients (52%) had a mutation in the *ENG* gene, 11 (44%) patients had a mutation in the *ACVLRL1* gene and in 1 patient (4%) with clinical HHT, no disease-causing mutation was found in the *ENG*, *ACVLR1* or *SMAD4* genes. Of the patients included for (mainly/ partly) epistaxis (*n* = 23), 21 (91%) patients had received previous surgical interventions and 21 patients (91%) had received systemic therapy for epistaxis. All the baseline characteristics are shown in Table 1. During the study, five patients (20%) did not complete the entire trial: two due to serious adverse events (SAE), two due to side effects and one for personal reasons. Thus, in total 20 patients (80%) were included in the final analyses. In Figure 2, the participant flow during the trial is shown. During the trial, three patients (15%) underwent surgical interventions: gastrointestinal argon plasma coagulation (APC) in two patients (one treated mainly for gastrointestinal bleeding, one treated for both epistaxis and gastrointestinal bleeding) and APC of the nasal mucosa in one patient (treated mainly for epistaxis).

### 3.2. Outcomes

#### 3.2.1. Pharmacokinetic Data

The starting dose of tacrolimus was 1 mg in 24 of the 25 patients, although we lowered this dose in one patient to 0.5 mg due to possible interaction with nifedipine, a calcium channel blocker, which can increase tacrolimus levels. During the trial, this patient ultimately required a dose of 1.5 mg tacrolimus daily to reach the target trough levels. The median tacrolimus dose at the end of the trial (week 24) was 2.0 mg daily (IQR 1.5–2.5 mg, range 1–5 mg). The median trough levels at the end of the trial were 2.5 μg/L (IQR 1.9–3.1 μg/L; range 1.6–4.3 μg/L). Figure 3 shows the trough levels of tacrolimus.

#### 3.2.2. Safety and Side-Effects

In 16 out of 25 patients (64%), at least one AE occurred during tacrolimus treatment. The majority of the AEs were mild and transient. The most frequently observed AEs were headache (*n* = 10), abdominal pain (*n* = 8), diarrhea (*n* = 8) and insomnia (*n* = 5). In five patients who finished the trial (25%), the AEs were still present. During the trial, we did not observe increased serum creatinine levels. The median serum creatinine levels at baseline were 71 (IQR 53–89) μmol/L and after 20 weeks of tacrolimus treatment, 71 (IQR 62–87) μmol/L. In one patient with moderate kidney disease, the creatinine levels were stable during the trial. One other patient with known elevated levels of alkalic phosphatase and gamma-GT (due to the presence of hepatic vascular malformations), had stable levels during the trial. In the other patients, no laboratory abnormalities were observed.

Two SAEs occurred during this trial. After 6.5 weeks of treatment, one patient had a neck abscess and subsequent S. Aureus bacteremia, which required several weeks of intravenous antibiotic treatment. Another patient was diagnosed with acute lymphatic leukemia after ten weeks of tacrolimus treatment and, shortly after diagnosis before the start of treatment for the leukemia, was infected with SARS CoV-2 and died due to respiratory failure. These were considered unrelated SAEs. Five individuals (20%) prematurely terminated the trial: four due to (S)AEs that occurred during tacrolimus treatment and one for personal reasons. Both the SAEs observed in this trial (neck abscess and acute leukemia) led to premature discontinuation. In addition, another patient stopped due to multiple AEs occurring within the first few days of treatment (headache, nausea, abdominal pain, dry mouth, gingivitis and increased urinary frequency) with a corresponding trough level < 1.0 µg/L; the other patient stopped both due to AEs (headache and insomnia) and due to little or no effect of tacrolimus on bleeding severity after 18.5 weeks of treatment. The indication of tacrolimus in all the patients who prematurely discontinued the trial was mainly epistaxis (5 out of 16 patients, 31%). All the (S)AEs observed during tacrolimus treatment are shown in Table 2.

#### 3.2.3. Hemoglobin and Ferritin Levels

At baseline, the hemoglobin levels were 6.1 (IQR 5.2–6.9) mmol/L (9.8 g/dL) and significantly increased to 6.7 (6.5–7.1) mmol/L (10.8 g/dL) at the end of the trial, *p* = 0.003. The hemoglobin levels did not significantly change in patients suffering from epistaxis alone (*n* = 11) or suffering from gastrointestinal blood loss alone (*n* = 2). In patients suffering from both epistaxis and gastrointestinal bleeding (*n* = 7), hemoglobin levels significantly increased from 5.5 (4.6–6.7) mmol/L (8.9 g/dL) at baseline to 6.8 (6.5–7.0) mmol/L (11.0 g/dL) at the end of the trial, *p* = 0.028. Additionally, patients received decreased packed red blood cell support throughout the trial, compared to the support prior to participation; see Section 3.2.6. Ferritin levels did not significantly change. In Figure 4, the hemoglobin levels at baseline and the end of the trial are shown. In Table 3, the outcomes are shown.

#### 3.2.4. Epistaxis Severity

The ESS decreased significantly in all patients (*p* = 0.003) and specifically in the subgroup of patients treated with tacrolimus because of epistaxis alone (*p* = 0.003). The ESS did not significantly change in the patient group with both gastrointestinal bleeding and epistaxis. The monthly epistaxis number and their duration at baseline compared to the end of the trial showed a pattern similar to that for ESS among the groups. The epistaxis severity is shown in Figure 5.

#### 3.2.5. Quality of Life, Fatigue and Experience of Treatment

The QoL measured with the PCS and MCS of the SF-36 did not significantly change. The fatigue complaints measured with the MFI-20 were also not significantly affected by tacrolimus treatment. However, 18 out of 19 patients (95%) were satisfied (10 patients very satisfied) with the treatment received, and 14 patients (74%) stated willingness to receive the treatment again.

#### 3.2.6. Iron Infusions and Blood Transfusions

In the 20 weeks before the start of treatment, 9 patients received a blood transfusion with a total of 99 transfusions; 2 patients received one transfusion, 2 patients, ten transfusions, 1 patient each 5, 6, 8, 23 and 35 transfusions. During tacrolimus treatment, 9 patients received blood transfusions; 1 patient one transfusion; 5 patients two transfusions; 1 patient four transfusions; 1 patient 11 and 1 patient 12 transfusions. The number of blood transfusions significantly decreased in all patients combined; non-parametric: prior to trial median 0.0 (IQR: 0.0–7.5; range: 0–35), during trial median 0.0 (IQR: 0.0–2.0; range: 0–12), *p* = 0.046 and parametric: prior to trial mean 5.0 (±9.2), during trial mean 1.9 (±3.5); *p* = 0.040.

During the study, 12 patients received intravenous iron. In the 20 weeks preceding tacrolimus use, 13 patients received intravenous iron. There was no significant change in the number of iron infusions needed before enrollment compared to that during the trial. The number of iron- and blood transfusions in the 5 months before tacrolimus and during the 5 months of tacrolimus treatment are shown in Table 4.

## 4. Discussion

This is the first trial investigating the effects of oral tacrolimus on HHT-related bleeding. We observed a significant increase in hemoglobin levels with a significant decrease in the number of blood transfusions needed and a decrease in epistaxis severity.

Besides the increased levels of ENG and ALK1, patients with HHT might benefit from other pharmacologic effects of tacrolimus. Tissue damage and subsequent inflammation are believed to play an important role in HHT pathophysiology. For example, inducing dermal wounds and lipopolysaccharides that trigger inflammation can also induce the formation of arteriovenous malformations in animal models of HHT [16]. In addition, in humans, there is evidence that trauma or inflammation contributes to the formation of abnormal vasculature [17]. Tacrolimus inhibits calcineurin and thereby inhibits T-cell function [18]. The immunosuppressive effect may block the inflammation that normally contributes to the formation of abnormal blood vessels in patients with HHT. Furthermore, elevated vascular endothelial growth factor (VEGF) plasma levels have been reported in patients with HHT [19], and VEGF inhibition decreases bleeding in patients with HHT [19,20]. There is evidence that tacrolimus also inhibits VEGF [21,22].

We observed a significant increase in hemoglobin levels, especially in the patients with HHT who suffered from both epistaxis and gastrointestinal bleeding (increase in hemoglobin of 1.3 mmol/L or 2.1 g/dL), while in the patients mainly treated with tacrolimus for epistaxis, the hemoglobin levels increased only slightly (0.4 mmol/L or 0.64 g/dL). Patients with both epistaxis and gastrointestinal bleeding may include those most severely affected (e.g., multiple organ involvement), and, therefore, the effect of tacrolimus treatment might be higher. However, interestingly, in the patients with both epistaxis and gastrointestinal bleeding, the epistaxis severity (measured by ESS and the epistaxis diary) did not significantly change during tacrolimus treatment. This may suggest that oral tacrolimus is more effective in decreasing gastrointestinal bleeding than epistaxis. On the other hand, patients with only epistaxis showed decreased ESS and epistaxis number and duration, which is also of clinical relevance. Definitive conclusions are difficult however, due to the small patient numbers. The timing of blood transfusions and iron infusions may have influenced the hemoglobin levels (both at baseline and the end of the trial). Also, the absence of a standardized management plan for the anemia may have influenced the results.

The QoL and fatigue did not significantly change after initiation of tacrolimus treatment. In our previous studies, we observed a significant increase in QoL in patients with epistaxis treated with oral itraconazole [23] and a significant decrease in fatigue in patients treated with octreotide [24]. It is possible that the negative effects caused by the adverse events (for example insomnia) could have counterbalanced the positive effects of tacrolimus on fatigue and QoL. Insomnia is a well-known side-effect of tacrolimus [25]. In this clinical trial, insomnia occurred in 5 out of the 25 patients (20%). On the other hand, the vast majority of the patients, 18 out of 19 patients (95%), were satisfied or very satisfied with the treatment received. Of note though, since this is a non-randomized open-label study, there is a higher risk of placebo effects. However, 12 out of the 20 patients that completed the trial (60%) restarted tacrolimus treatment after recurrence or increased HHT-associated bleeding, because of the beneficial effects during the study.

Approximately half of the participants suffered at least one AE. Fortunately, most frequent AEs in this study were mild. By contrast, in solid organ transplant recipients, common side effects are hypertension, headache, insomnia, nausea, diarrhea, impaired renal function and hyperkalemia, hyperglycemia (and diabetes mellitus). Two SAEs occurred: one patient suffered from a neck abscess and bacteremia, which might have been related to the immunosuppressive effects of tacrolimus. Another patient suffered from acute lymphatic leukemia and died from SARS CoV-2 infection. To our knowledge, no link between tacrolimus treatment and acute leukemia has been described in the literature. It is not likely that the immunosuppressive effects of tacrolimus influenced the outcome of SARS CoV-2 infection in this patient, since tacrolimus was discontinued directly after hospital admission for the leukemia. Also, the immunosuppressive effects of low dose oral tacrolimus are likely minimal because the target trough levels are substantially lower than in solid organ transplant recipients (2–3 μg/L versus 4–20 μg/L). In addition, in solid organ transplant recipients, other immunosuppressive drugs are usually combined with tacrolimus [26,27]. Therefore, both SAEs were considered as highly unlikely to be related to tacrolimus treatment in the trial according to the pharmacist, internist-immunologist and transplantation specialist consulted.

The strength of our study was the prospective design with a standardized protocol and predetermined outcomes. We assessed patient-reported outcome measurements such as change in QoL and fatigue symptoms. Our study is limited by the small sample size and the absence of a control group. HHT is still a rare disease, so inclusion of large patient groups is challenging. The small sample size reduces the chance of detecting a true effect and reduces the likelihood that a statistically significant result reflects a true effect [28]. These results are, nevertheless, strengthened by subgroup analyses which, despite the sample size being even smaller, does show significance. In this study, the lack of statistical significance does not automatically mean there is no effect at the individual patient level and the data must be interpreted carefully [29]. In addition, multiple comparisons performed in this study increase the likelihood of finding a false ‘significant’ association. Another limitation of this study is that we did not assess the cost-effectiveness of tacrolimus. Although tacrolimus is reasonably affordable, a formal cost-effectiveness study taking into account the costs of tacrolimus, the trough level measurements and the reduction in blood transfusions and hospital visits was not carried out since the study population was small, but it could, nevertheless, be very useful. Selection bias could have played a role, especially since no control group was present and, therefore, no comparison was possible. We believe, however, that this was limited by the fact that the St. Antonius Hospital is the only HHT center in the Netherlands and that the population studied reflects the HHT population suffering from severe bleeding caused by HHT [23].

A large, randomized, placebo-controlled trial designed to further assess the efficacy, safety and cost-effectiveness of tacrolimus would avoid these limitations. It would, however, need to be carried as an international trial so that more HHT centers could be included.

## 5. Conclusions

In conclusion, oral tacrolimus significantly improved hemoglobin levels and significantly decreased transfusion needs in patients with HHT and severe bleeding. Side effects of tacrolimus were common. The potential therapeutic benefit of oral tacrolimus and its safety should be further investigated in a randomized controlled clinical trial.

## Figures and Tables

**Figure 1 jcm-11-05280-f001:**
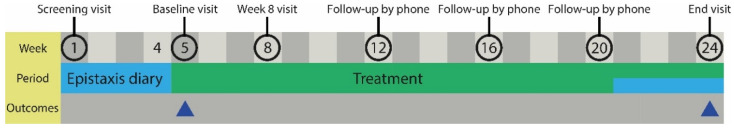
Schematic of the study schedule. The primary and secondary outcomes were measured at baseline and the end visit (blue arrowhead).

**Figure 2 jcm-11-05280-f002:**
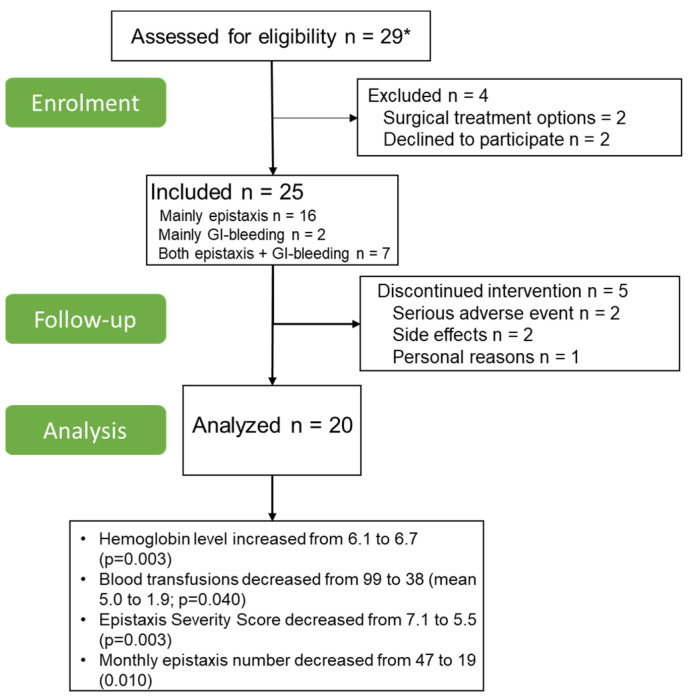
Participant flow during the trial. * Most patients were followed-up or referred to the pulmonologist for systemic therapy. The three pulmonologists specialized in HHT in our hospital were aware of the most important inclusion and exclusion criteria, and they had, therefore, already selected eligible patients.

**Figure 3 jcm-11-05280-f003:**
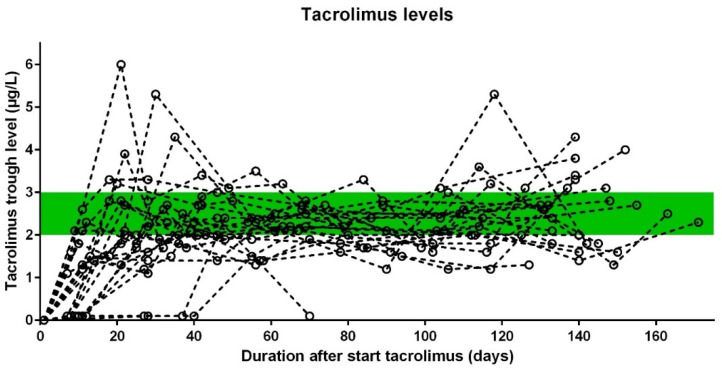
Tacrolimus trough levels for each patient.

**Figure 4 jcm-11-05280-f004:**
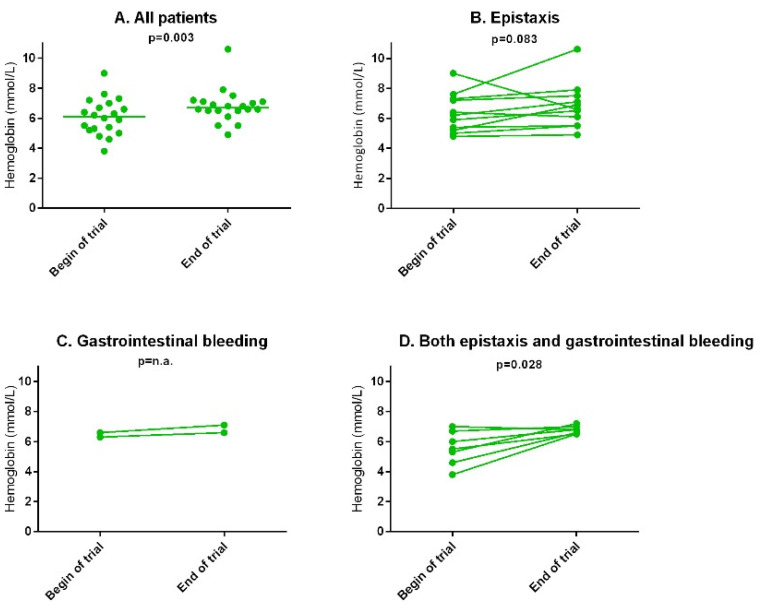
The primary outcome: median hemoglobin levels at baseline and the end of the trial. The bar is the median level. (**A**) For all patients. (**B**) For patients treated mainly for epistaxis. (**C**) For patients treated mainly for gastrointestinal bleeding. (**D**) For patients treated for both epistaxis and gastrointestinal bleeding.

**Figure 5 jcm-11-05280-f005:**
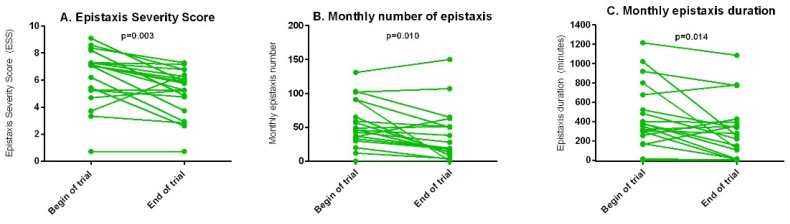
Epistaxis severity measurements at baseline and at the end of the trial. (**A**) Epistaxis Severity Score. (**B**) Number of epistaxis events monthly. (**C**) Monthly epistaxis duration in minutes.

**Table 1 jcm-11-05280-t001:** Baseline characteristics.

Parameter	Value
Number of patients, *n*	25
Median age, years (IQR)	59 (52–66)
Sex, *n* (%)	
Female	11 (44)
Male	14 (56)
Clinical diagnosis (≥3 Curaçao criteria), *n* (%)	21 (100)
HHT type, *n* (%)	
Type 1	13 (52)
Type 2	11 (44)
Type unknown	1 (4)
Visceral localization, *n* (%)	
PAVM on chest CT	13 (52)
CAVM *	1 (2.8)
Digestive tract telangiectases ¥	10 (40)
Previous systemic epistaxis treatment, *n* (%)	
Tranexamic acid oral	17 (71)
Propranolol	5 (21)
Bevacizumab	4 (17)
Previous surgical interventions for epistaxis, *n* (%)	
Argon Plasma Coagulation	22 (92)
Septodermoplasty	10 (42)
Nasal embolisation	4 (17)
Anticoagulant therapy before enrollment, *n* (%)	0 (0)
Smoker, *n* (%)	
Never	6 (24)
Former	11 (44)
Active	8 (32)
Median body mass index, kg/m^2^ (IQR)	25 (23–28)
Tacrolimus usage, *n* (%)	
Epistaxis	16 (64)
Gastrointestinal bleeding	2 (8)
Both epistaxis and gastrointestinal bleeding	7 (28)

CAVM, cerebral arteriovenous malformation; HHT, hereditary hemorrhagic telangiectasia; *n*, number; IQR, interquartile range; PAVM, pulmonary arteriovenous malformation; SD, standard deviation. * CAVM screening has been performed in 18 (72%) individuals. ¥ Digestive tract has been evaluated in 13 (52%) individuals.

**Table 2 jcm-11-05280-t002:** Adverse events according to the Common Terminology Criteria for Adverse Events (CTCAE) version 5.0.

Adverse Event	Number of Patients, *n*	Grade
Headache	10	Grade 1
Abdominal pain	8	Grade 1
Diarrhea	7	Grade 1
	1	Grade 2
Insomnia	5	Grade 1
Nausea	3	Grade 1
Muscle cramp	3	Grade 1
Tinnitus (increase in severity of pre-existing tinnitus)	2	Grade 1
Obstipation	1	Grade 1
Hyperhidrosis	1	Grade 1
Nasal congestion	1	Grade 1
Dizziness	1	Grade 1
Urinary tract infection	1	Grade 1
Urinary frequency (increase)	1	Grade 1
Dry mouth	1	Grade 1
Gingivitis	1	Grade 1
Skin infection (neck abscess and subsequent bacteremia)	1	Grade 3
Acute lymphatic leukemia, SARS CoV-2 infection	1	Grade 5

Grade 1 Mild; asymptomatic or mild symptoms; clinical or diagnostic observations only; intervention not indicated. Grade 2 Moderate; minimal, local or noninvasive intervention indicated; limiting age-appropriate instrumental daily activities. Grade 3 Severe or medically significant but not immediately life-threatening; hospitalization or prolongation of hospitalization indicated; disabling; limiting self-care daily activities. Grade 4 Life-threatening consequences; urgent intervention indicated. Grade 5 Death related to AE.

**Table 3 jcm-11-05280-t003:** Overview of the outcomes.

Parameter	Patients, *n*	Baseline	End of Trial	*p*-Value
Hemoglobin levels, mmol/L (IQR)	All patients (*n* = 20)	6.1 (5.2–6.9)	6.7 (6.5–7.1)	0.003 *
	Epistaxis (*n* = 11)	6.2 (5.2–7.3)	6.6 (5.5–7.5)	0.083
	GI bleeding (*n* = 2)	6.5	6.9	n.a.
	Both (*n* = 7)	5.5 (4.6–6.7)	6.8 (6.5–7.0)	0.028 *
Hemoglobin levels, g/dL (IQR)	All patients (*n* = 20)	9.8 (8.4–11.2)	10.8 (10.5–11.4)	0.003 *
	Epistaxis (*n* = 11)	10.0 (8.4–11.8)	10.6 (8.9–12.1)	0.083
	GI bleeding (*n* = 2)	10.4	11.0	n.a.
	Both (*n* = 7)	8.9 (7.4–10.8)	11.0 (10.5–11.3)	0.028 *
Ferritin levels, ug/L (IQR)	All patients (*n* = 20)	50.5 (13.5–283.5)	58.0 (16.3–426.3)	0.117
SF-36	All patients (*n* = 19) †			
PCS		35.1 (29.4–46.0)	39.2 (35.5–46.1)	0.260
MCS		46.0 (38.2–52.8)	48.4 (38.3–56.3)	0.117
Fatigue	All patients (*n* = 19) †			
General fatigue		17.0 (14.0–19.0)	16.0 (13.0–18.0)	0.325
Physical fatigue		16.0 (13.0–18.0)	15.0 (13.0–17.0)	0.419
Reduced activity		14.0 (13.0–16.0)	14.0 (12.0–16.0)	0.585
Reduced motivation		10.0 (8.0–15.0)	10.0 (8.0–13.0)	0.435
Mental fatigue		10.0 (7.0–16.0)	9.0 (6.0–14.0)	0.219
ESS (IQR)	All patients (*n* = 20)	7.1 (5.3–7.3)	5.5 (4.0–6.4)	0.003 *
	Epistaxis (*n* = 11)	7.3 (7.1–8.4)	6.1 (4.9–6.8)	0.003 *
	Both (*n* = 7)	5.3 (4.7–7.1)	5.3 (4.8–5.9)	0.463
Monthly epistaxis number, *n* (IQR) ^#^	All patients (*n* = 19) †	47 (33–91)	19 (9–51)	0.010 *
	Epistaxis (*n* = 10) †	46 (36–60)	19 (14–41)	0.005 *
	Both (*n* = 7)	97 (32−110)	63 (3–107)	0.600
Monthly epistaxis duration, minutes (IQR) ^#^	All patients (*n* = 19) †	377 (257–677)	253 (16–392)	0.014 *
	Epistaxis (*n* = 10) †	503 (371–946)	237 (118–456)	0.007 *
	Both (*n* = 7)	297 (172–400)	357 (16–426)	1.000

ESS, epistaxis severity score; GI, gastrointestinal; IQR, interquartile range; MSC, mental component summary; PCS, physical component summary; SF-36, Short Form Health Survey 36. ^#^ Indicates parameters that were measured in the one month before enrollment and during the last month of the trial. † One patient in the epistaxis group did not return the questionnaires and did not record the epistaxis diary at the end of the trial, thus, in total 19 were available for analysis in all patients and 10 in the epistaxis group. * Significant *p*-value.

**Table 4 jcm-11-05280-t004:** Outcomes: iron infusions and blood transfusions.

Parameter	Patients, *n*	5 Months before Enrollment	5 Months during Trial	*p*-Value
Iron infusions, number	All patients (*n* = 20)	38	35	
Iron infusions, median (IQR; Range)	All patients (*n* = 20)	1.0 (0.0–3.8; 0–6)	1.0 (0.0–3.8; 0–6)	0.499
Iron infusions, mean ± SD	All patients (*n* = 20)	1.9 ± 2.0	1.8 ± 2.0	0.505
Blood transfusions, number	All patients (*n* = 20)	99	38	
Blood transfusions, median (IQR, range)	All patients (*n* = 20)	0.0 (0.0–7.5; 0–35)	0.0 (0.0–2.0; 0–12)	0.046 *
	Epistaxis (*n* = 11)	0.0 (0.0–6.0; 0–10)	0.0 (0.0–2.0; 0–4)	0.207
	GI bleeding (*n* = 2)	Range: 10–35	Range: 0–12	n.a.
	Both (*n* = 7)	0.0 (0.0–5.0; 0–23)	2.0 (0.0–2.0; 0–11)	0.498
Blood transfusions, mean ± SD	All patients (*n* = 20)	5.0 ± 9.2	1.9 ± 3.5	0.040 *

GI, gastrointestinal; IQR, interquartile range; SD, standard deviation. * Significant *p*-value.

## Data Availability

The data presented in this study are available on request from the corresponding author. The data are not publicly available due to privacy reasons.

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
