# Peer review of "Efficacy and Safety of Tacrolimus as Treatment for Bleeding Caused by Hereditary Hemorrhagic Telangiectasia: An Open-Label, Pilot Study"

_jcm, 2022, doi:10.3390/jcm11185280_

Round 1

Reviewer 1 Report

Many agents have been tried in HHT to improve epistaxis/anemia.  Tacrolimus has been added to that list.  As with the others, it seems to be an incremental improvement - maybe.  I have many questions/concerns about this study.

Abstract:
Line 34: Need to specify transfusions over what period of time (5 months prior vs 5 months on treatment, for example)

Introduction:
Don't use "HHT patients" - people are more than their disease.
Line 49: HHT patients --> adults with HHT
Line 52: iron-deficient anemia --> iron-deficiency anemia
Line 52-52:  that requires --> that may require 
Line 53:  Moreover, the quality of life and fatigue of HHT patients...of the bleeding --> Moreover, quality of life and fatigue are often...bleeding in patients with HHT
Line 59:  How did you decide which agent (tacro vs siro) to test?
Line 73: toleration --> tolerability
*Even before I read on, I was super worried about placebo affect in your results! (non randomized open label)

Treatment Plan:
I am very concerned that it does not appear that safety labs were performed in your study (renal panel, hepatic profile, electrolytes) as this VERY IMPORTANT in the use of tacrolimus. In addition, as an immunosuppressant, should be following ANC and ALC as well, at least wbc.  Certainly you would have seen some abnormalities and these would have been shown in AEs had you done them.  This is a BIG BIG safety concern and I feel uncomfortable with the study for this reason.

Outcomes
Line 150:  contentment --> satisfaction

RESULTS
Line 179:  You mention "systemic therapies" here.  Would refer to Table 1 here.  Be careful to include only TXA oral, propranolol, and bevacizumab as the other treatments listed here are NOT systemic.
Line 194-195:  Would consider giving HGB in g/dL as this is commonly used.  I calculated the conversion and note that there was an average of 1g/dL improvement after 5 months of treatment.  yet these patients had 2 SAEs, 4 dropouts, 64% of AE.  Is that number even clinically significant?  Seems unlikely to me - they remain "mildly anemic" with HGB at baseline of 9.8, improving to 10.8 with treatment.
-In Table 2 - would * the significant p values in the right column
Line 218-219 and Table 2:  The most impressive change was in monthly epistaxis duration - the is statistically significant and it appears the decrease was also potentially CLINICALLY significant, with reduction ~50% bleeding time (and both lowest and highest numbers at baseline are down 50% as well).  I would consider an intrapatient graph showing each patient's improvement.  THIS is the one potential outcome that would change our patient's lives.
Line 224:  Very suspicious this is largely placebo effect.
Section 3.2.4:  Did patients stop taking after 5 months, or choose to continue on?  How long did the decrease in infusion/transfusion continue, how quickly was the effect lost?
Table 4:  Was the ALL considered a related SAE?  If so, wow.  Were patients notified of the death on study?  Notably absent here are abnormal electrolytes etc - suggesting to me that they were not collected and patients were put at unnecessary risk by not being appropriately monitored while on this trial.  Did you notice changes in BP?  Kidney function?  GLucose?
Paragraph starting 315:  I think you nail it here.  You EXPECTED improvement in QOL and fatigue and did not see.  Perhaps that means that tacro is just too toxic for this very limited very subtle not clearly clinically significant improvement in HGB.  I strongly suggest that you monitor more closely and that you focus on the things that seemed significant - change in blood/iron needs, change in epistaxis duration, rather than just HGB - IF you are able to move on to placebo-controlled phase 3 study.  I suspect it will be more difficult to enroll with "ALL" and "death" listed in possible outcomes.

While I appreciate the urgency to find things to improve the lives of our patients with HHT, it greatly concerns me that perhaps the appropriate safety labs were not performed, that the outcomes do not see very clinically significant, and that researchers continue to push drugs with high potential toxicity/low clinical improvement just for an incremental change.  It's time to be smarter, not automatically test every drug that has ever worked in another vascular anomaly.

Author Response

Dear editor,

Please see the attachment for the response to the reviewer. 

Reviewer 2 Report

OVERALL

This study is the first multi-patient clinical study assessing the potential safety and effective as of tactility’s as a for the treatment of HHT. It is a small, prospective trial, but reports several promising findings suggesting tacrolimus warrants further consideration.

 MINOR

- Authors report some significant improvements in Hb and ESS but not in all groups. We’re paired comparisons performed between measurements taken before and after the trial (Figure 2)?  If not, this may be helpful to illustrate how individual patients’ levels were affected by tacrolimus.

- Similarly, it would be helpful to see in graphical form how individual patients’ ESS scores changed before and after the trial.

- there is a formatting error on line 255-256.

-Authors may want to consider adding to the discussion a consideration of the side effects identified in this trial vs known side effects when tacrolimus is administered at higher dose for organ transplant.

Author Response

Dear editor,

Please see the attachment for the response to reviewer 2. 

Reviewer 3 Report

Review for Journal of Clinical Medicine

“Efficacy and safety of tacrolimus as treatment for bleeding caused by hereditary hemorrhagic telangiectasia: an open-label, pilot study”

This report describes a prospective, phase 2 open label study of low dose tacrolimus for the treatment of refractory anemia in HHT among patients at a single center in the Netherlands. 

This agent is of considerable interest as a potential therapy for HHT related bleeding as a consequence of studies that suggest an increase in expression of the deficient proteins (either ENG or ACVRL1) with tacrolimus, and also reports of improved bleeding among HHT affected recipients of solid organ transplants. Case reports have suggested benefit and objective, prospective data has been eagerly anticipated.

The report is generally well written (with only a couple of awkward wordings or typographical errors).  The results are mixed and will be of great interest to the HHT community.  The study naturally has limitations that are acknowledged and discussed.

SPECIFIC COMMENTS:

1.      Protocol and description.

a.      The inclusion criteria are unclear.  The study requires definite diagnosis of HHT (3+ Curaçao criteria or genetic diagnosis) appropriately.  However, the clinical features for inclusion are not clear, “They had anemia (sex-depended; hemoglobin level <7.3 mmol/L for females and <7.8 mmol/L for males), iron deficiency (ferritin level <10 µg/L) or had received iron treatment or blood transfusions in the last six months, and suffered from either at least 4 episodes of epistaxis per week and/or documented gastrointestinal telangiectases by endoscopy with suspicion of bleeding (e.g., melena, anemia disproportional to epistaxis).”  Was anemia alone sufficient?  Anemia with iron deficiency alone without any iron therapy?  Was iron treatment in the last six months alone sufficient without anemia?  Please provide a more unambiguous description of inclusion criteria. The inclusion criteria are vital to identifying a population with a certain level of risk that would justify use of a medication with potential serious adverse events.

b.      The exclusion criteria are described much less ambiguously.

c.       The manuscript would benefit from a CONSORT diagram outlining the population screened for the study, those who failed screening including the number excluded for each exclusion criterion, those who additionally refused to consent, and those randomized.  This would provide a better sense of the proportion of HHT patients eligible and willing to assume this protocol/therapy.  The CONSORT diagram should outline the patients that failed to complete the protocol (and the reasons for it), and the population that were included in the analysis.  The authors could consider including the primary outcome results in the same diagram for a single concise visual representation of the trial.

d.      As the SI units for hemoglobin are not universally adopted or understood, and since hemoglobin is key for the primary outcome, please also provide the equivalent hemoglobin ranges and results in g/dL or mg/dL throughout the manuscript.

e.      The anemia/iron management protocol should be discussed with greater detail.  The primary outcome (change in hemoglobin level between baseline and following treatment) would be influenced by bleeding severity, but also would be heavily influenced by iron therapy and transfusion threshold.  A standard anemia management plan for these patients would have established uniform decision making for the physicians managing these patients, yet this study left anemia management to non-study physicians without clear guidelines for thresholds to invoke therapy (and these physicians were not blinded to patient participation in the trial).  If the local practice of these physicians is/was fairly uniform, a description of that practice pattern would help with interpretation of the results. Fortunately, the rise in hemoglobin did not seem to require a greater intensity of blood or iron infusions (as appropriately outlined).  This actually strengthens the argument that tacrolimus has beneficial effects on bleeding.

2.      Results

a.      It is a significant concern that serious adverse events occurred in 8% of the trial participants, including a death that may have been related to immunosuppression.  Both safety and efficacy are important endpoints in considering a new therapy, but before considering the efficacy of the study among the 80% of the initial participants that did not withdraw, there should first be a discussion of the drug exposure, the withdrawal rate and the adverse events (ie: sections 3.2.5 and 3.2.6 should be discussed first, before sections 3.2.1-3.2.4).

b.      As acknowledged by the authors, there is considerable risk of beta error due to small sample sizes, and alpha error (check) due to multiple comparisons.  In general I feel that the authors used appropriate language in this regard.

c.       The epistaxis diary is important data to suggest efficacy of the therapy.  Improvement was seen in epistaxis frequency and total duration.  Total duration is the result of the product of nosebleed frequency and duration.  However, the calculated mean duration of individual nosebleeds appears to have increased (8.0 minutes/bleed in the run-in period, and 13.3 minutes/bleed on treatment – cannot comment on statistical significance without reviewing the source data).  Since total epistaxis duration is contaminated by epistaxis frequency, the frequency should be reported per bleed, with statistical assessment.

d.      I would use caution in interpreting the subjective assessment of patients contentment with the trial and willingness to be treated again.  As an open label study, there is considerable risk of placebo effect, especially for subjective outcomes such as quality of life and patient satisfaction.  The more objective measures of quality of life and fatigue were not improved in the study, and thus the improvement in patient satisfaction and willingness to repeat treatment is less impressive, and likely highly influenced by placebo effect.

3.      Minor language concerns

a.      Section 2.5 “…mean and standard DEVIATION (SD)…”

b.      Section 3.2.6 “…we did not observe and increase in serum CREATININE levels.  The median serum CREATININE levels…”

c.       Line 349 “…reduction in blood transfusions and DAYCARE visits was not carried out…”. There is likely a difference in the use of the term daycare in Dutch and English, I’m not entirely sure what this means.

Author Response

Dear editor,

Please see the attachment for the response to reviewer 3. 

Round 2

Reviewer 1 Report

This version is dramatically improved from prior.  I appreciate the edits as well as the additional information about the trial.  This is now an excellent paper.

I have just a few remaining comments/suggestions for readability.

Line 92:  sex-depended --> sex-dependent
Line 95: suffer from --> experience
Line 96: at least 4 episodes --> 4 or more episodes
Line 124: who were treated --> who had been treated
Line 125: before this trial --> prior to this trial
Line 193:  is showed --> is shown
Line 289: One patient --> In one patient
Line 290: Another one patient --> One other patient
Line 299: Those were considered --> These were considered
Line 352: only suffered from epistaxis --> suffered from epistaxis alone
Line 352: or only suffered from GI blood loss --> or GI blood loss alone

Line 357:  Would consider adding "although patients received decreased packed red  blood cell support throughout the trial than prior to participation, see section 3.2.6."  This is important (and impressive).

Line 404:  enrolment --> enrollment
Table 3:  Please add * to the signfiicant p-values as in Table 2.

Line 465:  probably notably ess than usual --> likely decreased as compared to the literature
Line 469: non-related --> unrelated

Line 471-473:  Would move this sentence elsewhere.  Maybe to line 456 after "suffered at least one AE."  Then change: Fortunately, most common AEs were mild --> Fortunately, most common AEs in this study were mild

Author Response

Dear editor,

Please see the attachment for the responses to the reviewer. 
